# Assessing Delivery of Selected Public Health Operations via Essential Public Health Operation Framework

**DOI:** 10.3390/ijerph17176435

**Published:** 2020-09-03

**Authors:** Mariam Safi, Maja L. Bertram, Gabriel Gulis

**Affiliations:** 1Department of Medicine Sønderborg/Tønder, Hospital of Southern Jutland Denmark, DK-6400 Sønderborg, Denmark; 2Unit for Health Promotion Research, University of Southern Denmark, Niels Bohrsvej 9-10, 6700 Esbjerg, Denmark; mbertram@health.sdu.dk (M.L.B.); GGulis@health.sdu.dk (G.G.)

**Keywords:** essential public health operation (EPHO), health services, organization of health services, local government

## Abstract

Background: To assess the conduct of delivery of public health services at the municipal level in Denmark by applying services enlisted in the Essential Public Health Operation framework (EPHO) of WHO. Methods: We conducted individual qualitative interviews with key informants working with public health using a self-assessment survey tool in order to obtain an in-depth understanding of the interrelation or multidisciplinary work in Kolding Municipality. The developed self-assessment survey tool entailed questions about essential public health activities performed in a municipality. Results: The Municipality organizes and contributes to core service delivery EPHOs, namely health protection, health promotion, and disease prevention. It collaborates with the general practitioners and the Region of Southern Denmark, responsible for hospital care, to fulfill the selected EPHOs. Conclusions: To obtain a comprehensive picture of the organizations that deliver public health services within a municipality, it is necessary to conduct interviews with representatives from those organizations as well. Additionally, the results from this study can be used to improve the survey tool further and hereafter conduct a nationwide survey in Denmark, as well as other European countries.

## 1. Introduction

The public health discipline has in the last 50 years developed rapidly and contributed significantly to the improvement in population health [1]. The rising burden of the non-communicable diseases (NCDs) [2], ageing population [3], increasing health inequalities, and healthcare costs [4] underlines the importance of investing in public health and creates a need for comprehensive public health services.

The European Health 2020 Policy tackles the aforementioned challenges, presenting new rationally goal-oriented public health strategies for promoting health [5]. The Member States adopted the European Action Plan for Strengthening Public Health Capacities and Services (EAP-PHS), which provides the framework for implementation of Health 2020 [6]. Central to the EAP-PHS is the list of the ten Essential Public Health Operations (EPHOs) [6], which is based on the list of ten Essential Public Health Services, developed by the Centers for Disease Control and Prevention (CDC) in 1994 in the USA [7]. The CDC was the first organization that defined public health services and provided a guideline for responsibilities of local public health systems, as well as performance indicators [7]. The list of ten EPHOs reflects on the public health landscape in Europe [8]. The ten EPHOs are divided into three sub-groups: EPHO 1 + 2 are the intelligence-oriented services, EPHO 3 + 4 + 5 are the core public health services, and EPHO 6–10 are the supportive services. The delivery of public health services varies across the European Region because of the differing national priorities, demographics, financial resources, etc. The list of EPHOs describes the elements within and outside the public health system that are expected to work together to provide high-quality public health services to the population. In addition, the EPHOs are applied as a measurement tool to analyze public health systems or organizations at the national or local governmental level [8]. However, according to the midterm progress report on the EAP-PHS [9], its potential has remained unrealized. It is apparent that many EU Member States have still not utilized or fully acknowledged the EPHOs, and this poses as a barrier in the process [9]. The Member States are still slow in the process of establishing comprehensive public health services compared to the USA [10].

Very few studies have analyzed or described how public health systems are organized, financed, and evaluated in different European countries. Allin et al. [11] described public health decision-making systems, and the book “Facets of Public Health in Europe” [12] described public health in Europe, from the aspect of different functions. According to the “Public Health Capacity in the EU Final Report” [10], assessing or measuring public health capacity is a challenging task indeed. One of the main challenges is obtaining data, because there are different understandings of public health services among the European countries [10]. Recent findings from the midterm progress report on the EAP-PHS [9] suggested that many key public health actors in their countries were, in fact, unaware of the EAP-PHS and the EPHOs. Hence, the EPHOs have limited involvement in public health planning within public health organizations. Further, the lack of financial resources, political commitment, evidence, and collaboration between health organizations are barriers hindering the implementation of the EAP-PHS. These issues highlight the need for committed leaders within health organizations that endorse the EPHOs and act on the results from the EPHO self-assessment tool [6].

In Denmark, the Danish Health Act 2005 (Sundhedsloven) [13] delegated a range of public health-related responsibilities to municipalities. Yet, to our knowledge, no study has assessed the organizational systems of delivery of public health services at the local level in Denmark by applying the EPHO framework [10]. Due to the organization of the public health system in Denmark [14], we focused on services described under EPHO 3, 4, and 5 [8]. A brief summary of the Danish health system based on “Health systems in Transition; Denmark (https://www.euro.who.int/__data/assets/pdf_file/0004/160519/e96442.pdf?ua=1) is available in Appendix B of this manuscript. These services do not include direct medical services.

The primary aim of this pilot study was to assess organizational systems and the coordination of delivery of three core EPHOs (health protection, health promotion, and disease prevention) in a Danish municipality by applying criteria and services enlisted in the EPHO framework of WHO. We selected Kolding Municipality to study the organization of public health services. Kolding is located on the Eastern part of the Jutland peninsula and, with a population of about 93,175 (larger municipality), it is one of the biggest Danish municipalities; the core city of Kolding has about 61,121 inhabitants. Kolding Municipality was selected due to its size and convenience in regard to access, location, and time. The specific objectives of the present study were:To develop a survey tool that fits the Danish system and apply it in Kolding Municipality.To identify who is providing core service delivery EPHOs at the municipal level in Denmark.

## 2. Method

To obtain an in-depth understanding of the interrelation or multidisciplinary work in Kolding Municipality, it was optimal to conduct individual qualitative interviews with key informants working with public health using a self-assessment survey tool (Appendix A). This form of self-assessment via individual interviews was most suitable to our context as the likelihood of conducting and completing the self-assessment was high in the presence of the interviewer.

A self-assessment survey tool based on the selected EPHOs (3, 4, and 5) was developed; it is based on a set of 20 Indicators of Local Health System Performance developed by Mays [15]. The guiding criteria laid in the EPHO self-assessment tool was used as well when developing the survey tool. The purpose of the instrument was to collect information about the essential public health activities performed in the municipality and what type of organization or actors contributed to these activities. The tool was based on key areas such as organizational structure, partnerships, health promotion, and protection. In addition, it was important to adjust the survey tool to the Danish system and the selected EPHOs (3, 4, and 5).

The survey consisted of 17 questions, which sufficiently covered the important aspects of public health activities in local communities in Denmark. The survey began with a brief profile description of the participant, which included their contact information, profession, the unit they worked in, and information about the duration of their employment in Kolding Municipality. This is followed by the organizational questions asking the participants whether a specific activity was performed in their jurisdiction. We translated the survey into Danish.

Two individual interviews were conducted: One with a health consultant within the Health and Training unit and the other with the Chief of Health and Training unit (senior administrative). However, conducting the survey in an interview format allowed us to collect in-depth data information and gave us feedback on what needed to be revised in the survey tool such as: Amending ambiguous phrasings and terms; tailoring the questions according to a health or non-health unit; not asking questions about law and regulations, because the responsibility lies within the government.

### Sample Selection and Data Collection

A copy of the survey tool was sent to a health consultant within Kolding Municipality to identify key participants accordingly. Ideal participants were senior administrators within the health unit who had general knowledge about the types of public health activities, performed by their agencies and by other organizations within the community.

In collaboration with the health consultant, four senior administrators and one health consultant were identified within the Health and Training unit and invited for individual interviews. The health consultant accepted the invitation and worked in the field of alcohol and rehabilitation. Out of the 4 senior administrators, only one accepted the invitation on behalf of all of them. It was particularly challenging to access informants. One key reason for this was the lack of time and resources on their side. However, the two informants were viable as we obtained in-depth information about the essential public health activities (EPHOs) performed by Kolding Municipality. The senior administrator was the chief of the Health and Training unit. The tasks of the Chief of Health and Training are to develop, coordinate, and implement health promotion initiatives, including creating frameworks for equal access to health through development and interdisciplinary cooperation in the areas of healthy food, smoking, alcohol, exercise, and mental health.

Steps in the qualitative analysis included:Preliminary exploration of the data by reading through the transcript and writing memos;Sorting the data based on the EPHOs (3, 4 and 5) and their sub-operations [16].

## 3. Results

The following section presents the findings on how Kolding Municipality organizes and contributes to core service delivery EPHOs, as well as the contribution and participation of other agencies or health organizations. Figure 1 presents the institutions and organizations involved in delivery of public health services.

The findings showed that Kolding municipality is responsible for part of the services described in the core EPHOs only.

Within Kolding Municipality, the Health and Training unit is mainly responsible for providing health services in the areas of nutrition, smoking, alcohol, physical activity, and mental health. Within chronic diseases, prevention efforts are aimed at addressing diabetes, cardiovascular diseases (CVD), osteoporosis, and chronic obstructive pulmonary disease (COPD). The municipality is not responsible for public health services such as food protection, occupational health protection, immunization programs, and screenings programs. This is aligned with the Danish Health Act 2005 (Sundhedsloven) [13], which outlines the overall responsibilities of the municipalities.

In the following, we present services where Kolding Municipality has full or partial responsibility for provision of core EPHOs.

### 3.1. EPHO 3

#### 3.1.1. Environmental Health Protection

Within Kolding Municipality, the City and Development Department is responsible for environmental health including water, soil, housing, and occupational health, and has an environmental policy framework, and planning and housing guidelines. The policy framework and guidelines are developed based on the established directives from the Food and Environmental Protection Agency. However, the Health and Training unit is not responsible for environmental health.

#### 3.1.2. Occupational Health Protection

The Health and Training unit offers workplace health programs to public and private corporations. The aim is to promote and support healthy choices and hereby create an enabling environment for healthy behaviors among workers. However, the Health and Training unit is not responsible for the legislative framework for occupational health.

Occupational health services in Denmark are primarily regulated by the National Working Environment Authority.

### 3.2. EPHO 4

#### 3.2.1. Intersectoral and Interdisciplinary Capacity

The municipality establishes their health policy and health plan based on the overall guideline provided by the Ministry of Health. It is the Association of municipalities (KL) and Association of Danish Regions (Danske Regioner) who specify and apply the national guidelines.

Kolding Municipality is also part of Sundhedsstrategisk Forum (Health Strategy forum), which is a meeting forum for all the municipalities in the Region of Southern Denmark. It is a platform where representatives from the municipalities, hospitals, and general practitioners (GPs) meet regularly to support, discuss, and formulate public health strategies. The forum addresses key health issues such as cancer, diabetes, cardiovascular diseases, musculoskeletal disorders, and mental health, and establishes health plans that describe the responsibilities of the municipalities, hospitals, and GPs in the specific health areas.

The municipality also has a communication network with the National Health Board, the National Serum Institute (SSI), and the National Institute of Public Health (SIF).

#### 3.2.2. Health Promotion: Behavioral and Social Risk Factors

The Health and Training unit develops, coordinates, and implements health promotion, and coordinates efforts carried out by different teams. They ensure that the citizens have equal access to healthcare through interdisciplinary cooperation in the areas of nutrition, smoking, alcohol, exercise, and mental health. Within chronic diseases, prevention efforts are aimed at diabetes, CVD, osteoporosis, and COPD. The municipality collaborates with the Region of Southern Denmark and the Danish Regions and conducts a national health profile survey “Hvordan har du det? (How are you?),” which looks at contributing factors (nutrition, tobacco, alcohol, and physical activity (KRAM –factors from Danish “Kost-Rygning-Alkohol-Motion”)) to prioritize health needs.

In addition, Kolding Municipality participates in a nationwide youth profile survey conducted by Senior- and Social Department and Police (SSP-Network). It provides a descriptive profile of changes in the youth development related to health, crime, drugs, education, and working or leisure time.

Moreover, the municipality has timely communication with GPs regarding health promotion and prevention programs including counseling and support programs to patients at the healthcare center. Likewise, Kolding Municipality cooperates with the hospitals about care and rehabilitation that do not take place during hospitalizations.

The Health and Training unit of Kolding municipality coordinates intersectoral health promotion and prevention activities as well. They engage in health promotion programs with, e.g., Senior Unit, Job Center Kolding, and Abuse Center. Every department or team has various collaborations (intern or extern) with different organizations when working on certain projects. Central to these partnerships are that the organizations are either directly or indirectly tied together.

### 3.3. EPHO 5

#### 3.3.1. Primary Prevention

The main responsibility of primary prevention rests with the GPs, the hospitals, and the SSI. The SSI is responsible for surveillance and control of communicable diseases, while the GPs and hospitals are obliged to report instances of certain communicable diseases to the SSI. The GPs delivers public health services related to vaccination and immunization programs. The municipality including Health and Training is not responsible for primary prevention programs, and the responsibilities lie within another jurisdiction.

#### 3.3.2. Secondary Prevention

The primary care sector, hospitals, is mainly responsible for secondary prevention services such as screening programs for early detection of diseases and maternal child health programs. However, Kolding Municipality delivers maternal and child health services. It includes educational and management programs for first-time mothers, post-delivery, and infants care programs.

#### 3.3.3. Tertiary/Quaternary Prevention

It is Kolding Municipality’s responsibility to provide rehabilitation and chronic pain management programs to patients before and after hospitalization. Furthermore, the municipality oversees nursing homes and provides care at homes for the elderly. The senior department is mainly responsible for elderly care. Support and educational programs for marginalized and vulnerable groups are present.

Figure 1 roughly depicts how Kolding Municipality organizes and contributes to core service delivery EPHOs, as well as the contribution and participation of other agencies or health organizations.

## 4. Discussion

The findings from the survey showed that Kolding Municipality is only partially responsible for services described under three core EPHOs, which is fully in line with § 119 of Danish Health Act, nr. 546 from 24 June 2005 [13]. The municipality organizes and contributes to service delivery of EPHOs with the cooperation of internal sectors and other health and non-health organizations. As internal within the municipality, the Health and Training unit is primarily responsible for health promotion and disease prevention. In addition, the Health and Training unit of the municipality collaborates with the City development unit to ensure limited provision of environmental health-related services. The study also elucidated that the health sector in Kolding Municipality works closely together with the GPs and the Region of Southern Denmark, which is responsible for the hospitals, when coordinating health plans. To obtain a full picture about organizations and institutions involved in the conduct of three core EPHOs, a detailed mapping of each organization would be necessary including all relevant services described under EPHOs. Moreover, to draw a full picture of the public health system, such an analysis would need to cover all 10 EPHOs on the local, regional, and national level and describe connections and coordination mechanisms.

A major challenge with such a mapping exercise is to construct a reliable and comprehensive mapping/survey tool. The WHO Self-assessment tool [8] contains a long set of questions to assess the conduct of services under each EPHO, yet the focus is on the services itself and not on organizations providing the service. We attempted to shorten the list of services under each EPHO and keep the completeness high, yet participants expressed that the survey was too long and time-consuming to complete. Inspired by the survey tool of Glen P. Mays, translated by three co-authors of this manuscript, the two native Danish speakers developed the used version, which was considered a pilot version. Therefore, formal rules for validated translation were not followed, creating a potential limitation to the validity of the results.

Another important issue is selection of participants; in informal discussions with participants, it was mentioned that the survey tool requires quite a comprehensive knowledge of public health and EPHOs as such. Therefore, a high level of expertise is needed to complete the survey. This knowledge criterion would gain even more importance if the survey addresses non-health departments as well. Due to the intersectorality approach, inclusions of, for example, environmental, or social affair units would be highly requirable in future work.

Additionally, the participants (health leader and health consultant) were unaware of the EPHOs, which posed as a barrier. Recent findings from the midterm report on the EAP-PHS also highlight that many key public health actors do not have knowledge of the EPHOs, which makes the assessment and implementation of the EPHOs challenging indeed [9].

Despite all limitations, this was the first pilot study that investigated how a local public health organization in Denmark organized and contributed to core service delivery EPHOs, as well as the contribution and participation of other agencies or health organizations. A strength of the present study was that the survey was conducted in face-to-face interviews, which enabled us to collect in-depth information. This also allowed the interviewer to clarify questions that the participant found confusing or ambiguous and, therefore, provided feedback on what needed to be improved in the self-assessment survey tool.

This pilot study can be used as a framework to conduct a nationwide survey in Denmark related to core service delivery EPHOs. The findings from this study can be used to refine the survey tool so it is even more specific to the Danish system. As many of the sub-operations in the core service delivery EPHOs belong to non-health units, it is necessary to include those units in the future to obtain a more comprehensive picture of how delivery of services is organized. The “ideal respondent” is a senior administrative who has in-depth knowledge of the types of public health activities performed by their municipality and by other agencies within the community. To enhance clarity in organization of public health systems responsible for the conduct of EPHOs, similar surveys would be recommended to other countries.

## 5. Conclusions

In the present study, a survey tool, based on the set of 20 Indicators of Local Health System Performance by Glen P. Mays and the guiding criteria laid in the EPHO self-assessment tool, was developed. The core function of the survey instrument was to assess the essential public health activities performed within a municipality in Demark and the types of organization or key actors that contributed to these activities. The developed survey tool was tested using Kolding Municipality, the Health and Training unit, as a setting. The findings showed that Kolding Municipality organizes and contributes to core service delivery EPHOs at limited extent and in cooperation with internal sectors and other health and non-health organizations. Due to the existence and use of the WHO public health service self-assessment tool, it was not the aim of our study to assess the quality of provision of public health services [8]. Despite the conduct of the self-assessment in many countries of Europe [9], identification, description, and assessment of organizational types of public health services, especially within a municipal setting, are important to improve the health planning and delivery of these services.

More research is needed in the area of public health system research in order to draw a comprehensive picture of delivery of public health services within a municipal level in Denmark and internationally. A detailed description of provision of individual public health services in terms of organizations should allow for better harmonization and cooperation among different units or institutions involved in the work. Although other national health systems might need different tools related to the issues of system governance, going in depth in EPHO 6 was not objective of this paper, the pilot study can be useful for assessing the delivery and coordination of public health services within other Danish-relevant governance settings.

The findings from this pilot study can be useful for assessing the delivery and coordination of public health services within other relevant governance settings [16,17].

## Figures and Tables

**Figure 1 ijerph-17-06435-f001:**
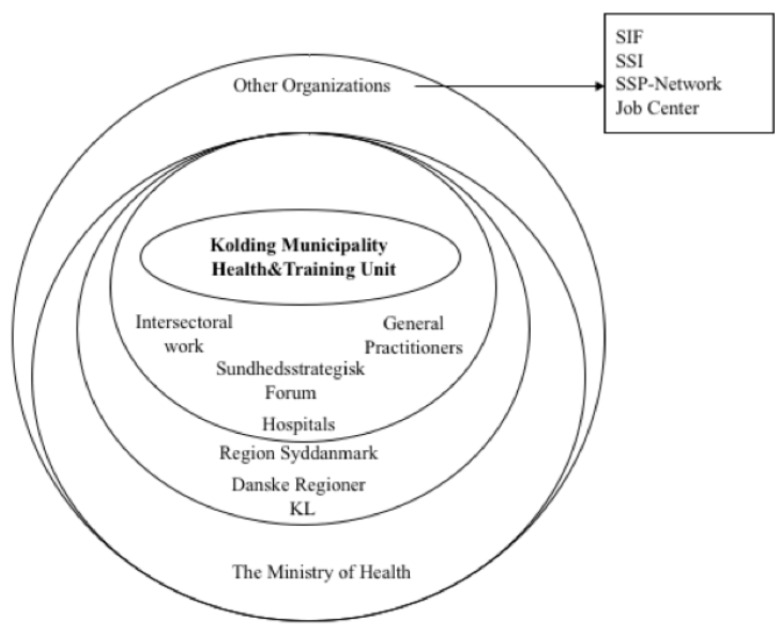
Kolding Municipality’s partnerships.

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
