# Peer review of "Assessing Delivery of Selected Public Health Operations via Essential Public Health Operation Framework"

_ijerph, 2020, doi:10.3390/ijerph17176435_

Round 1

Reviewer 1 Report

This paper has tried to assess how the Kolding Municipality performs against European criteria for the delivery of public health services. This type of assessment of public health services is done far less frequently than, for example clinical services.

The stated aim of the pilot study was to assess the presence of organisational systems and the delivery of core public health programs in health protection, health promotion and disease prevention.

There were some inherent difficulties in doing so, the most important of which is that this municipal organisation has direct responsibility for only some of these functions. The authors pointed out that they sought senior informants for face to face interviews for this study.

A key issue for this type of research is of course, do public health organisations have the capability to deliver these services. The conclusion to the paper is weak - stating only that this municipality could contribute to the performance of these functions. This seems to be damning by faint  praise. Some discussion of the difficulties of assessing the quality as well as the presence of services would strengthen this paper.

Author Response

Dear reviewer,

thank you for your general comment! We added a sentence into Conclusion explaining in principle why we do not focus on quality and capability to deliver the EPHO’s. There is an existing self-assessment toolkit for such work developed by WHO and used until now on national level in about 1/3rd of countries of European Region. Our interest was (and still is) to get a picture who is doing what on provision of EPHO’s especially on local level due to fact that in Denmark some services are delegated to local level. We found some positive facts, but also some missing knowledge. Therefore, our aim is to focus attention toward a need to describe in depth the organizational part of provision of services to allow for better harmonization and cooperation. 

Reviewer 2 Report

See attached archive

Author Response

Dear reviewer,

thank you for your comments. Please see our responses below.

The objective of this paper is to assess public health services at municipal level in Kolding Municipality (Denmark). Two individual interviews are developed.In my opinion, the collected data are very scarce and they do not guarantee the validity of the results. This problem is enough to reject this article.

Response: We agree with this limitation but not with the suggestion to reject. We are very aware that findings are valid for Kolding municipality and discuss this issue in Discussion. However, this is a pilot study aiming to focus attention toward conduct of similar studies and as product, provides a survey tool. The tool itself can be used for other Danish municipalities and later, after cultural adjustment in other countries. 

Moreover, the authors should explain the following shortcomings:

1) In the introduction, the authors refer to a list of ten Essential Public Health Operations (EPHOs). However, they select three of them without explaining why they focus on this three. An explanation about this choice should be included in the paper. Moreover, a more detailed explanation of the ten EPHOs would be convenient.

Response: We added a sentence explaining clearly why we selected EPHO 3, 4 and 5 in Introduction. There is no space to describe all ten EPHO’s and it could only lead to plagiarism if we would do so; we provide references, where people can read about them in detail.

2) International readers do not know whether the public health system of Kolding Municipality is similar to other Danish Municipalities or not. It is important this question in order to know whether the results are applied for the rest of Denmark.

Response: We added a new reference describing the healthcare system in Denmark where readers can find information on what belongs to municipalities. Yes, in terms of competences and responsibilities it is the same to each municipality in Denmark given by the law (reference in text). But we acknowledge the possible organizational differences in our Discussion when discussing validity of findings.

3) In the present version of the paper, it is not clear the utility of obtained results. The authors identify how Kolding Municipality organizes some public services but they do not provide information about the level of efficacy and efficiency of this organization. Do interviewed health consultant and senior Administrator think that this organization is optimal? How can it be improved? Are public health resources used appropriately?...

Response: we added a sentence on this issue in Conclusion. Our aim was not to assess efficacy, efficiency or quality; that can be done (and in many countries of European region already completed) by using the WHO self-assessment tool (reference provided in text). Furthermore, this is a major topic in itself that needs to be addressed separately and in-depth in order to give it the justice it needs. To include these topics in the paper, it will be also required to interview the municipal council which consist of politicians.Our aim was to identify organizational system of provision of public health services, even more specifically to develop a survey tool for such assessment and pilot test it in one municipality.

4) The survey questions should be included in an Annex.

Response: We provide it as supplementary material now.

Reviewer 3 Report

This is so interesting study about essential public health service. From the background of your paper, it can be seen that you have a good knowledge of essential public health, and may help other readers know the importance of your work and give guideline for further study.

1) Please check your Keywords's punctuationï¼›

2) In line 85, I suggest that it is better if you could add questionares as supplementary materials;

3) Line 139, Why you use EPHO 3, not EPHO 1?

4) In terms of part abount Environmental health protection, Occupational health protection, it is better to if you could expain in a more detail way. In addition, because both of them are not health service bureau, so you need more words to explain the relationship among different bureaus as providing public health servieces, eg, is medical health servieces or not, like this;

5)In EPHO 5, you classified health services according three different stages in preventive medicine. I guess maybe Environmental health protection, Occupational health protection can also be included in this catalog. So you need explain why you classsify like that, or just according to the EPHO?

6) Most public health services are taken by health services bureau, so you need discusss the role and extent (proportion, etc) of health services bureau in the process of providing the essential public health service.

7) You need give the full name of abbreviation when you use abbreviation.

8) There are many directory levels. Paper's format maybe need to be revised. 

Author Response

Dear reviewer,

thank you for your comments. Please see below our responses.

This is so interesting study about essential public health service. From the background of your paper, it can be seen that you have a good knowledge of essential public health and may help other readers know the importance of your work and give guideline for further study.

1) Please check your Keywords’ punctuationï¼›

Response: done, we deleted the semicolon

2) In line 85, I suggest that it is better if you could add questionnaires as supplementary materials;

Response: the questionnaire is added as supplementary material now as suggested by reviewer

3) Line 139, Why you use EPHO 3, not EPHO 1?

Response: we focused on core EPHO’s only as the Danish municipalities have no responsibility on intelligence related EPHO’s (EPHO 1+2)

4) In terms of part about Environmental health protection, Occupational health protection, it is better to if you could explain in a more detail way. In addition, because both of them are not health service bureau, so you need more words to explain the relationship among different bureaus as providing public health services, e.g., is medical health services or not, like this;

Response: we added a short description and a reference to a document describing the organization of healthcare in Denmark. The system is briefly explained also in method part where the respondent selection process is descried. We made it clear, that our target was not medical health services.

5)In EPHO 5, you classified health services according three different stages in preventive medicine. I guess maybe Environmental health protection, Occupational health protection can also be included in this catalogue. So, you need explain why you classify like that, or just according to the EPHO?

Response: This was not our choice; we used the categorization as described in WHO documents describing the EPHO’s.

6) Most public health services are taken by health services bureau, so you need discuss the role and extent (proportion, etc) of health services bureau in the process of providing the essential public health service.

Response: We do discuss the work done by Health & Training Unit in the municipality, as well as signalize links to other Units within and outside of the municipality (see figure 1 and the first paragraph in Discussion part. There is no health service bureau in Danish system; all is organized within Units of the municipalities. We also mention in discussion that to get a full picture, a detailed mapping of all Units of municipality would be necessary. 

7) You need give the full name of abbreviation when you use abbreviation.

Response: We believe all abbreviations are explained now when first used.

8) There are many directory levels. Paper's format maybe need to be revised. 

Response: The levels were due to attempt to provide the original coding as provided in WHO documents on services under each individual EPHO. We dropped those numbers and simplified the levels in Results part making it hopefully more understandable. 

Round 2

Reviewer 2 Report

I have read the new version of the paper. I remain of the same opinion that the collected data are not enough to give support to the results. 

Author Response

Dear Reviewer,

thank you for your short comment and responses in the template. We understand your opinion originating in low number of interviews and empirical data collected, yet we cannot agree with your summary conclusion. Let us first address your grading in the template.

I would like to repeat our research aim; we aimed to develop a questionnaire to map who is performing the individual public health operations as defined by EPHO's and pilot test that questionnaire in one Danish municipality. The choice of Denmark is given by our affiliation, and even more importantly by Danish public health system (reference provided in revised version of the manuscript).

We would be glad to improve the Introduction but, as the two other reviewers acknowledged there is a shortage of research on organizational and institutional systems performing public health functions, especially non-medical public health functions and the EPHO's. We are confident that we included all recent literature on this subject.

For declared aim of the study we considered of course a quantitative design and aimed to include more employees of selected municipality. However, it became quickly clear that people are not extensively familiar with EPHO's and therefore a larger sample was considered unreliable and we choose the qualitative design with key informants. In our view this was the best option and changing the Methods at this stage would mean a brand new study. We are planning such a study including whole Denmark, but that will be subject of a new manuscript. 

In terms of clarity of Results, we edited the opening part of the chapter with aim to make the presentation of results more clear. Thank you for this comment; we realized that this part of the text was a bit confusing and figure 1 was misplaced.

We edited the Conclusions with aim to make the link between the presented work and conclusions even more clear.

In your summary statement, you wrote "the collected data are not enough to give support to the results." We believe, by saying that you mean the issue of low sample size and we accept that criticism. However, we have to argue the other way around and we do so in the Discussion part of the paper. We are very aware of this weakness and therefore we clearly state that results are not generalizable. In Conclusion we directly invite to more similar research addressing organizational and institutional system of provision of EPHOs. In terms of this paper, we present and discuss only data received via interviews; it is not possible to present data not collected in results. 
